# Impact of Hydrological Modellers' Decisions and Attitude on the Performance of a Calibrated Conceptual Catchment Model: Results from a 'Modelling Contest'

**Helge Bormann** [1,*]**, Mariana Madruga de Brito** [2]**, Despoina Charchousi** [3]**, Dimitris Chatzistratis** [4]**,**
**Amrei David** [5]**, Paula Farina Grosser** [6]**, Jenny Kebschull** [1]**, Alexandros Konis** [7]**, Paschalis Koutalakis** [8]**,**
**Alkistis Korali** [8]**, Naomi Krauzig** [9]**, Jessica Meier** [10]**, Varvara Meliadou** [11]**, Markus Meinhardt** [12]**,**
**Kieran Munnelly** [2]**, Christiane Stephan** [2]**, Leon Frederik de Vos** [13]**, Jörg Dietrich** [14]
**and Ourania Tzoraki** [8]

[1] Department of Research and Knowledge Transfer, Jade University of Applied Sciences, Ofener Straße 16/19, 26121 Oldenburg, Germany; jenny.kebschull@jade-hs.de

[2] Department of Geography, University of Bonn, Meckenheimer Allee 166, 53115 Bonn, Germany; mariana@uni-bonn.de (M.M.d.B.); Kieran_Munnelly@hotmail.co.uk (K.M.); cstephan@uni-bonn.de (C.S.)

[3] NTU Athens, School of Rural and Surveying Engineering, Zografou Campus 9, Iroon Polytechniou str, 15780 Zografou, Greece; charchousi@gmail.com

[4] Department of Physical Geography, Faculty of Geosciences, Utrecht University, 3512 JE Utrecht, The Netherlands; d.chatzistratis@hotmail.com

[5] TU Darmstadt, Franziska-Braun-Str. 7, 64287 Darmstadt, Germany; a.david@ihwb.tu-darmstadt.de

[6] BTU Cottbus, Postfach 101344, 03013 Cottbus, Germany; paulagrosser@yahoo.de

[7] TU Dresden, Mommsenstraße 13, 01602 Dresden, Germany; alexkonis@hotmail.com

[8] University of the Aegean, University Hill, 81100 Mytilene, Lesvos, Greece; koutalakis_p@yahoo.gr (P.K.); alkistikorali@yahoo.gr (A.K.); rania.tzoraki@aegean.gr (O.T.)

[9] Parthenope University of Naples, Via Ammiraglio Ferdinando Acton, 38, 80133 Napoli, NA, Italy; naomi.krauzig@uniparthenope.it

[10] University of Potsdam, Am Neuen Palais 10, 14469 Potsdam, Germany; jessica.meier91@yahoo.de

[11] Democritus University of Thrace, University Campus, 69100 Komotini, Greece; v.meliadou@gmail.com

[12] University of Jena, Löbdergraben 32, 07743 Jena, Germany; markus.meinhardt@uni-jena.de

[13] TU München, Arcisstraße 21, 80333 München, Germany; frederikdervos@gmail.com

[14] University of Hannover, Appelstr. 9a, 30167 Hannover, Germany; Dietrich@iww.uni-hannover.de

**\*** Correspondence: helge.bormann@jade-hs.de; Tel.: +49-441-7708-3775

**Abstract:** In this study, 17 hydrologists with different experience in hydrological modelling applied the same conceptual catchment model (HBV) to a Greek catchment, using identical data and model code. Calibration was performed manually. Subsequently, the modellers were asked for their experience, their calibration strategy, and whether they enjoyed the exercise. The exercise revealed that there is considerable modellers' uncertainty even among the experienced modellers. It seemed to be equally important whether the modellers followed a good calibration strategy, and whether they enjoyed modelling. The exercise confirmed previous studies about the benefit of model ensembles: Different combinations of the simulation results (median, mean) outperformed the individual model simulations, while filtering the simulations even improved the quality of the model ensembles. Modellers' experience, decisions, and attitude, therefore, have an impact on the hydrological model application and should be considered as part of hydrological modelling uncertainty.

**Keywords:** conceptual hydrological model; HBV; modellers' decisions; model ensemble; modellers' uncertainty

## 1. Introduction

Reliable flood prediction and water management requires robust hydrological model applications. While a high accuracy of the prediction is usually desired, the underlying uncertainty is a major issue of predicting hydrological variables. Uncertainty analysis is, therefore, a key technique to assess the robustness of model simulations and to communicate the findings to decision makers.

Different sources of uncertainty are usually discussed and considered in the literature. Uncertainty in modelling [1] is usually attributed to:

(1) Structural and technical uncertainty caused by the underlying model philosophy, the representation of the hydrological processes, its degree of physics [2,3], and the computational implementation;
(2) parameter uncertainty [4,5]; and
(3) data uncertainty [6,7].

The international hydrological community has focused on these uncertainty sources for the past decades, however, researchers have only rarely considered the modeller her-/himself explicitly as a behavioural aspect of model uncertainty in environmental modelling [8]. Impacts of the modellers were mostly subsumed as part of the parameter uncertainty. However, the modeller takes decisions during the model selection and application process and has an impact on the model results.

What kind of decisions does a modeller take during model application? She/he chooses a model, selects adequate (or at least available) data, parameterizes the model, and selects objective functions and methods for calibration (e.g., manual vs. automatic optimization of parameters, structured sensitivity analysis prior to calibration). Modellers may have an incomplete understanding of the model, may not be fully motivated, or may have a lack of time to reach the best possible solution. Only few studies have tried to analyse such an impact of the modeller. The influence of modellers' decisions on model results in risk assessment was investigated by [9]. A study by [10–12] focused on the influence modellers have on a priori predictions of catchment hydrological processes by comparing a priori model applications to a small, artificial catchment in Northeast Germany. However, they struggled with the instance that modellers applied different models on the same catchment and data set. Thus, only the combination of modellers' decisions and the model used could be analysed. Typically, instead of searching for the perfect model for a distinct model application, most modellers prefer using their own and accordingly their standard model for model application varies in different catchments and for different purposes. This is due to trust and experience, and may include slight model adjustments depending on the application. This happened also in another study [10,11], and restricted them to analyse the combination of different models and modellers.

Model ensembles are often constructed to frame the uncertainty of a simulation. Early applications in numerical weather forecasting were done by perturbing the initial conditions of the model [13]. Other sources of uncertainty, or even combined sources of uncertainty, were used to generate ensembles. It was found that multi-model ensembles outperformed single-model ensembles [14]. Multi-model ensembles outperformed single models [15,16]. Ensembles were also successfully applied in flood forecasting [17]. However, analyses of single-model ensembles did not explicitly consider the impact of the modellers themselves. However, there is evidence of a collective intelligence of groups of humans, which can outperform the average intelligence and even the highest intelligence of single humans [18].

To overcome the shortcomings described, and to quantify the impact of the modeller on model application, we designed a modelling-contest-experiment as part of a DAAD (German Academic Exchange Service) summer school on floods and flood risk management. We asked different modellers with diverse hydrological modelling experience and attitudes to apply the same catchment model to one Greek catchment based on identical data and model codes. We assume that experience, attitudes, and the individual calibration strategy have an impact on the model results achieved by different modellers. Based on this experimental design, the power of model ensembles with regard to the impact of modellers' decisions was also investigated. Due to collective intelligence, we assume modellers'

ensembles to be superior to single modellers' results. Based on the conceptual design of the study (Section 2), we contribute to the following research questions:

(1) Do different modellers achieve different model results by applying the same model to the same catchment? (Section 3.1)
(2) Can the variability in the model results be explained by the experience, attitudes, and calibration strategy of the different modellers? (Section 3.3)
(3) Do building modellers' ensembles improve the model performance compared to the results achieved by individual modellers? (Section 3.4)

## 2. Materials and Methods

### 2.1. HBV Model

The HBV model (Hydrologiska Byråns Vattenbalansavdelning) is a spatially semi-distributed and conceptual hydrological catchment model [19,20]. In this study the HBV-light version of HBV is used [21]. HBV considers the division of various components, such as snow, soil, upper and lower reservoir, and the river, and estimates the water mass balance in each of them. A basin can be delineated in numerous sub-basins in relation to the user preferences, experience, and basin characteristics. For each sub-basin, HBV calculates the catchment discharge, usually on a daily time step, which is estimated based on time series of precipitation and air temperature as well as potential evaporation estimates. Time series of crop-water-need (Kc), the volume of water needed by the various crops to grow optimally, are provided by the user for each sub-basin. Groundwater recharge and actual evaporation are simulated as functions of the actual soil water storage. Snow accumulation and snowmelt are computed by a degree-day method. The routing routine is based on a triangular weighting function to simulate the routing of the runoff to the catchment outlet. Detailed model descriptions and the central equations can be found, for example, in [19–21].

A set of parameters (Table 1) must be calibrated by the user related to the snow routine, soil routine, upper and lower zone response routine, and the transformation routing routine, as well as the initial values in each reservoir. In previous studies, HBV type models were successfully applied to semi-arid catchments [22,23]. It can therefore be assumed that HBV can represent the dominant processes of Mediterranean catchments.

**Table 1.** Main calibration parameters of the HBV model.

| Model Parameters | Relevance | Indicative Value |
|:---:|:---:|:---:|
| x | Weighting factor related to the streams dimensions | 0.25 |
| k | Storage coefficient related to the stream dimensions | 0.55 |
| HL1/Z | Parameter related to the elevation | 30.00–63.00 |
| K0 | | 0.45–10.50 |
| K1 | Recession rate of the various reservoirs parameters | 10.70–20.70 |
| K2 | | 30.5 |
| PERC | The maximum percolation rate from the upper to the lower groundwater box | 1.00 |
| MAXBAS | Routing parameter of the triangular weighting function | 5.00–10.00 |
| FC | Maximum soil moisture storage | 800–1000 |
| BETA | Soil parameter | 2.00–10.00 |

## 2.2. Tsiknias River Catchment

The Tsiknias river catchment is located in the north-central part of Lesvos Island (Greece, Figure 1). The river drains an area of about 90 km². The elevation of the catchment ranges from 0 to 968 m. The topography is characterized partly as lowland and as middle-mountainous. The semi-arid climate of the area is typically Mediterranean, characterized by hot and dry summers and mild rainy winters with a high relative humidity. Two meteorological stations operate in the basin: (a) Stipsi (396 m altitude) with an annual mean rain value 870 mm, and (b) Agia Paraskevi (95 m altitude) with an annual mean rain value 664 mm. Average annual potential evapotranspiration is estimated around 1050 mm/a. Since 2014, a telemetric station (Prini) operates on the main channel, providing water level data at 15 min intervals. The main part of the watershed is covered by cultivated land (mainly olive groves), grassland, and brushwood. In a few source areas, pine and oak woods have developed. In most parts of the catchment, soils are developed from volcanic rocks. Soil permeability is low in most of the area, low-medium in the highland areas, and medium in the north part of the area near the estuary. Since the Tsiknias River frequently experiences flash floods, a robust model application is an essential requirement for regional flood prediction and management.

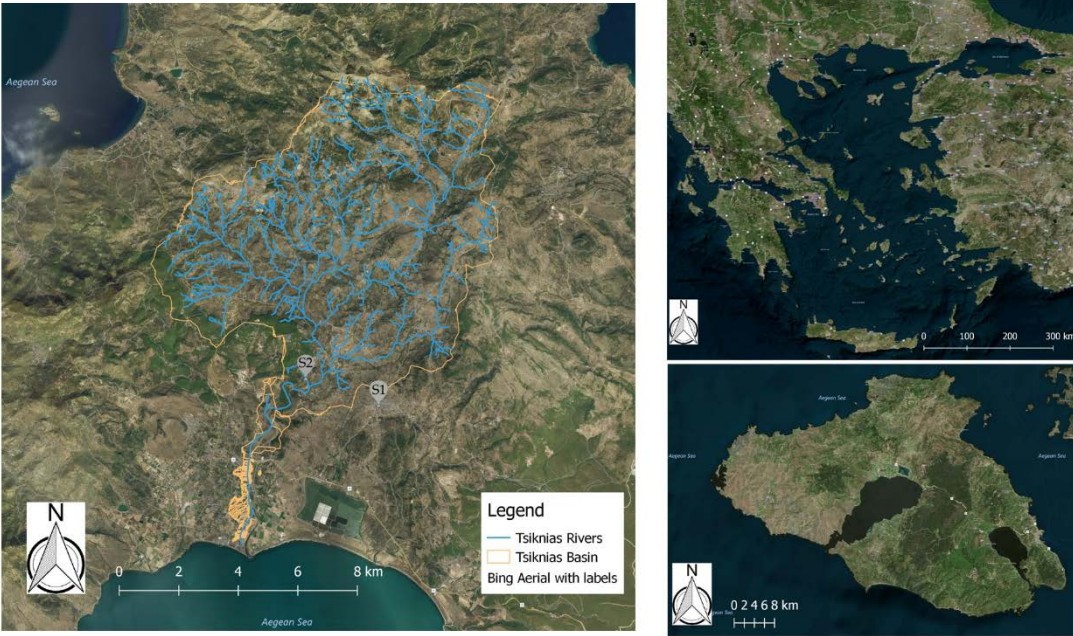

**Figure 1.** The Tsiknias River catchment (**Left**; upper **Right**: Greece; lower right: Island of Lesvos).

For this study, three years of daily data of the river gauge (Prini; July 2014 to March 2017) was used as well as daily data of one climatic station close to the river gauge. Data was quality-checked before model application. While usually more than three years of data should be used for hydrological catchment model application to represent hydrological variability, in the case of the Tsiknias catchment for this time period, the high quality of the data could be guaranteed. Starting the simulation period after the dry summer period, a warm-up period was not required assuming low antecedent moisture conditions.

## 2.3. Design of the Modelling Contest

After a general introduction into the HBV model theory, including an explanation of the process equations and the parameterisation of the model, all 17 participants of the modelling contest had one working day available for HBV model calibration to the Tsiknias River. MSc students, PhD students, and Postdocs with diverse modelling experience were participating in the experiment. All modellers were supplied with identical hydro-climatological data and model codes. They were asked not to

communicate on their individual model applications during the model contest. At the end of the day, modellers were asked to deliver their manually calibrated parameter sets and their simulation results, and to complete a questionnaire on their hydrological modelling experience and attitudes (see Section 2.4). The best hydrographs according to the Nash-Sutcliffe model efficiency [24] were collected from all modellers as well as their questionnaires. Both sources were used for further evaluation of the results.

### 2.4. Design of the Questionnaire

The questionnaire on the modellers' expertise and attitudes consisted of seven questions on modelling experience (based on [11]): If any, how many years, how many different models, experience with HBV (if yes, how many years), experience in semi-arid climates, experience in different climates), two questions on the calibration strategy (trial and error or sensitivity analysis or any other; choice of statistical quality measures for calibration), and a final statement whether the modellers enjoyed the exercise or not.

To analyse the results of the survey in a semi-quantitative way, the following classifications were assumed concerning modelling experience and attitudes:

(1)  Modelling experience was classified into three different classes: No experience, little experience (1 model, max. 1 year experience), some experience (either more than one model, or more than one year experience, or both); and

(2)  The degree of enjoyment was directly taken from the survey (yes/no); however, some participants replied that they enjoyed the contest in general, but they mentioned some criticism ("yes, but..."). They were ranked as a third class between "enjoyed" and "did not enjoy" for the evaluation ("mostly").

## 3. Results and Discussion

### 3.1. HBV Model Application to Tsiknias River Catchment

All participating modellers calibrated the HBV model manually to the Tsiknias catchment for a three year period. Figure 2 gives an impression on the variability of the best simulated hydrograph of each modeller, selected by maximum Nash-Sutcliffe efficiency (NSE). The variability among the simulated hydrographs is quite high, and both underestimation and overestimation of the observations during the rainy season were delivered by the modellers. Figure 3 emphasizes this variability in simulated hydrographs for three exemplary runoff events in early 2015.

Analysing the flow duration curves derived from observations and simulations revealed that none of the modellers succeeded in simulating the low flows during the dry season (Figure 4). While the modellers simulated "no flow" for 40–80% of the year, at least marginal flows were observed for the entire simulation period (discharge of less than 20 l/s for 31% of the year). The variability in simulated discharges was also high with regard to the simulated long term water balances (see Table 2).

**Table 2.** Statistical analysis of the goodness-of-fit criteria of the best simulations of each modeller.

| Goodness of Fit Index | Best Simulation | 75%-Percentile | 50%-Percentile | 25%-Percentile | Worst Simulation |
|---|---|---|---|---|---|
| Nash-Stutcliffe efficiency (NSE) | 0.56 | 0.53 | 0.47 | 0.35 | −45.9 |
| Coefficient of determination | 0.56 | 0.55 | 0.51 | 0.42 | 0.082 |
| Percent bias (PB) | 3.5% | 14.6% | 24.9% | 31.2% | 526% |
| Root mean squared error (RMSE) | 0.67 mm/d | 0.69 mm/d | 0.73 mm/d | 0.81 mm/d | 6.85 mm/d |

The absolute bias (PB) of the simulated discharge ranged from 3.5% (of the best simulation) up to 526% (of the worst simulation).

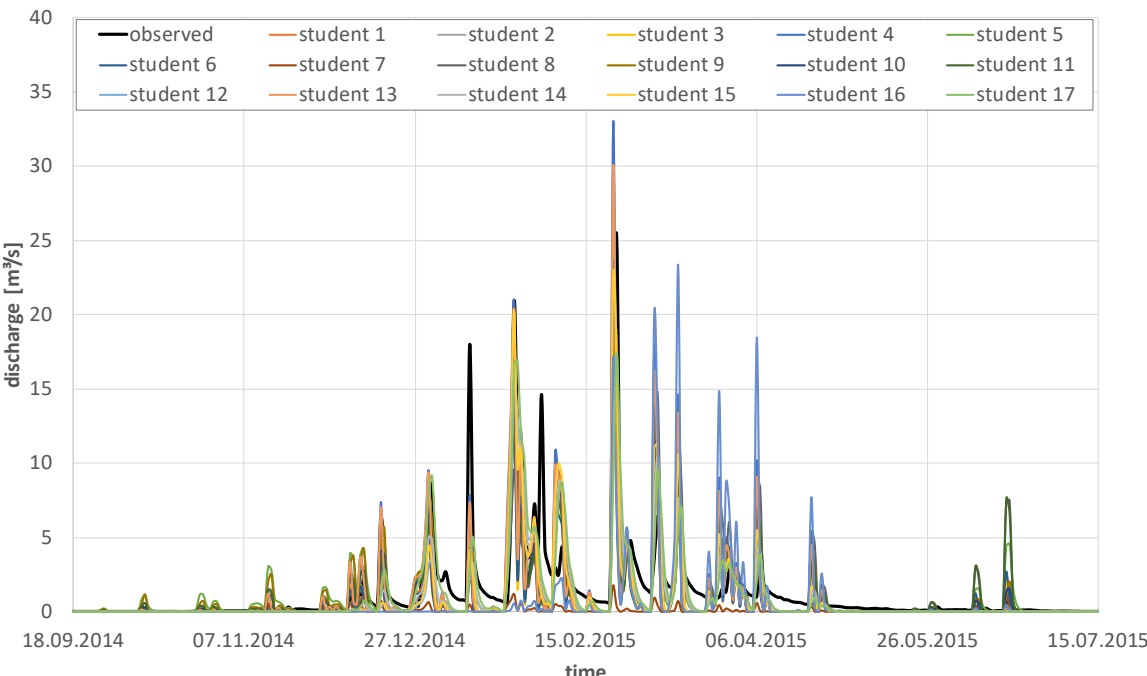

**Figure 2.** Variability of the discharges simulated by the different modellers against observations for an exemplary wet season (2014/2015) of the simulation period.

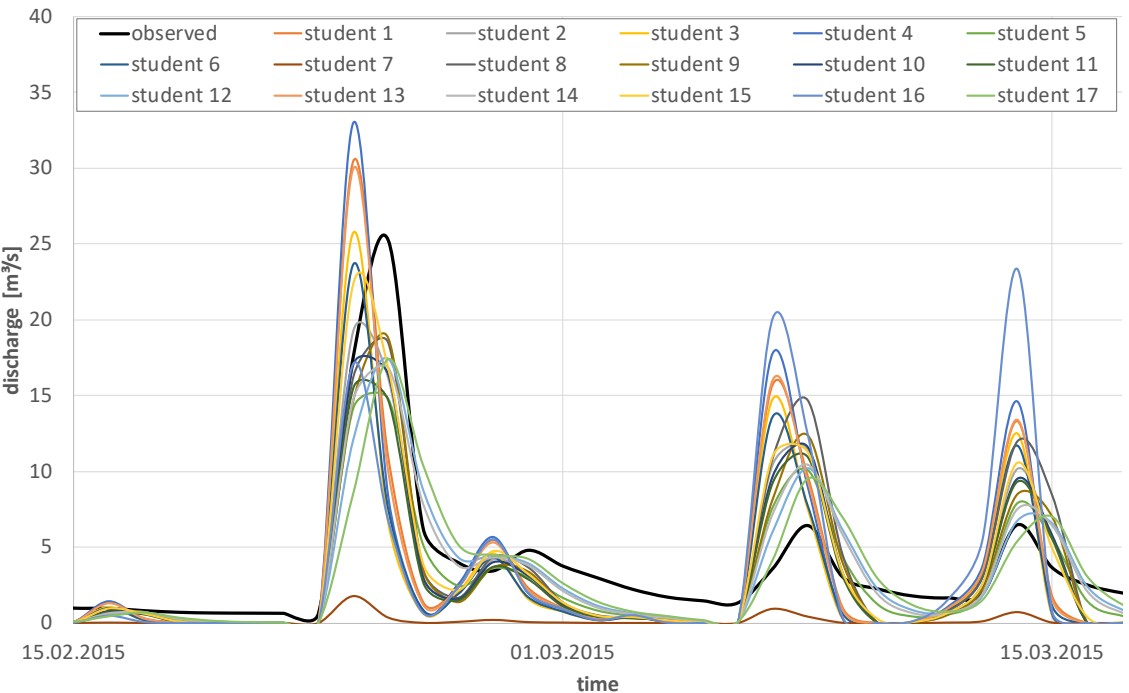

**Figure 3.** Variability of the discharges simulated by the different modellers against observations for three selected events in the rainy season of 2015.

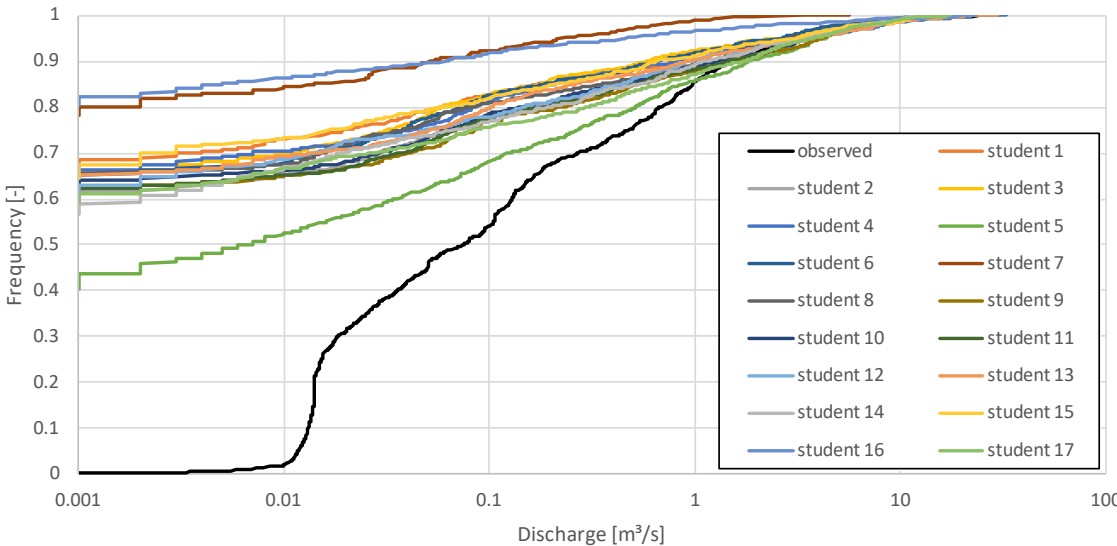

**Figure 4.** Flow duration curve based on observed (black) and simulated discharge for the whole simulation period.

While modellers used individual objective functions for calibration, a standard set of criteria was used for the evaluation of the goodness-of-fit of the model results. Statistical analysis of the goodness-of-fit criteria used for evaluation of the simulation results revealed that the simulation quality of the hydrographs delivered by the different modellers differed a lot.

Analysing the calibrated parameters sets did not yield distinct systematics. Depending on their individual calibration strategies, different modellers tried to optimize different parameter sub-sets, in exceptional cases also resulting in implausible parameters (see Table 3 for ranges of calibrated parameters). Since different parameter combinations resulted in similar goodness-of-fit values, parameter equifinalty is obvious for the selected parameter range.

**Table 3.** Parameter ranges of calibrated parameters of the HBV model.

| Model Parameters | Minimum Value | Median Value | Maximum Value |
|:---:|:---:|:---:|:---:|
| HL1/Z | 0.0004 | 0,037 | 65 |
| K0 | 0.0095 | 0.2 | 18 |
| K1 | 0.002 | 0.2 | 3.6 |
| K2 | 0.001 | 0.1 | 1 |
| PERC | 0.001 | 0.02 | 3 |
| MAXBAS | 4 | 7 | 98 |
| FC | 150 | 250 | 1140 |
| BETA | 1 | 2 | 10 |

The comparison against the simulation criteria defined by [25] yielded that eight out of 17 modellers achieved satisfactory results related to the Nash-Sutcliffe efficiency ([24]; NSE $\geq$ 0.5) while nine out of 17 modellers achieved good or satisfactory results related to percent bias (PB $\leq$ 25%).

These numbers showed that about half of the modellers achieved "satisfactory" simulation results (applying criteria that were designed for monthly time steps to daily simulations) while the remaining half did not (for details see Table 2).

The goodness-of-fit of the simulated discharges was relatively low compared to other HBV applications. We assume that this was at least partly due to the design of the modelling contest, being based on the limited time of the contest (one working day) and working in an unknown environment. Beyond that, we assume that hydrological simulation of a semi-arid catchment, being characterized by an intermittent flow regime, is a specific challenge.

*3.2. Experience, Decisions, and Attitudes of the Modellers*

The evaluation of the questionnaire revealed that the hydrological modelling experience of the participants differed a lot.

The following answers were provided on the modelling experience: While four out of 17 participants had no hydrological modelling experience at all, nine out of 17 participants had experience with one or two models, and four out of 17 participants had experience with three or more models. Almost half of the participants, eight out of 17, had less than one year modelling experience. Five out of 17 participants had one to two years modelling experience, and four out of 17 participants had three or more years modelling experience. Only one of the 17 participants had modelling experience with HBV. Beyond the modelling experience, knowledge on the regionally important characteristics and processes were asked for as well. Only five of 17 participants worked already in semi-arid regions while 12 out of 17 participants had never worked on the hydrology of semi-arid regions.

With regard to model calibration, most of the modellers calibrated their model by trial and error (10 out of 17) while only six out of 17 applied a systematic sensitivity analysis in order to reduce the number of calibration parameters selected for manual calibration. Most of the modellers used the NSE for calibration (14 out of 17) while only a few modellers considered the water balance (seven out of 17) or the root mean squared error (RMSE) (four out of 17). The use of multiple statistical quality measures did not guarantee achieving satisfactory modelling results. Eight of 13 modellers achieved satisfactory NSE according to [25]. However, those who did not consider more than one index, did not achieve satisfactory NSE values at all. Three of them achieved the smallest NSE values of all modellers. The analysis of the relation of modellers' experience to the calibration strategy and the choice of the quality measures used for calibration did not show any significant pattern.

Finally, 12 out of 17 participants enjoyed the modelling contest, while two out of 17 participants did not, and a further three out of 17 participants raised arguments pro and contra.

*3.3. Evaluation of Model Results against Modellers Experience, Decisions, and Attitudes*

The modelling results and the results of the survey were matched and evaluated against the NSE, which was selected for calibration by most of the modellers. Due to the limited number of modellers, the calibration strategy, attitudes, and experience of the modellers was grouped in few classes (Section 2.4) and individually compared to the simulation results. Comparisons between the classified groups are presented as follows. Due to the relatively small number of modellers, statistical significance of differences between different groups was not tested.

Relating the modelling experience against the goodness-of-fit revealed that an increasing experience helped to achieve a good modelling result while also, without any modelling experience, it was possible to achieve satisfactory modelling results according to [25]. However, the best NSE values, ranging from 0.53 to 0.56, were achieved by experienced modellers (Figure 5, left). Fifty-seven percent of the experienced modellers achieved better results than any of the less experienced modellers, and 71% of the experienced modellers achieved satisfactory simulation results (NSE $\geq$ 0.5) while the proportion decreased to 30% of less experienced modellers.

A similar picture was shown by analysing the experience on hydrological processes in intermittent catchments. While 60% of the experienced participants achieved satisfactory simulation results, the proportion of the unexperienced ones was significantly lower with 41%. Additionally, the variability within the simulation results of experienced modellers was lower compared to the unexperienced ones (Figure 5, right).

The comparison of the calibration strategies of the different modellers against the achieved NSE revealed large differences between the different options. From those participants who decided to apply a systematic sensitivity analysis, only one modeller did not achieve satisfactory results, while all others did (86%). From those modellers who did the calibration based on trial and error, only two (20%) achieved satisfactory results (Figure 6, left).

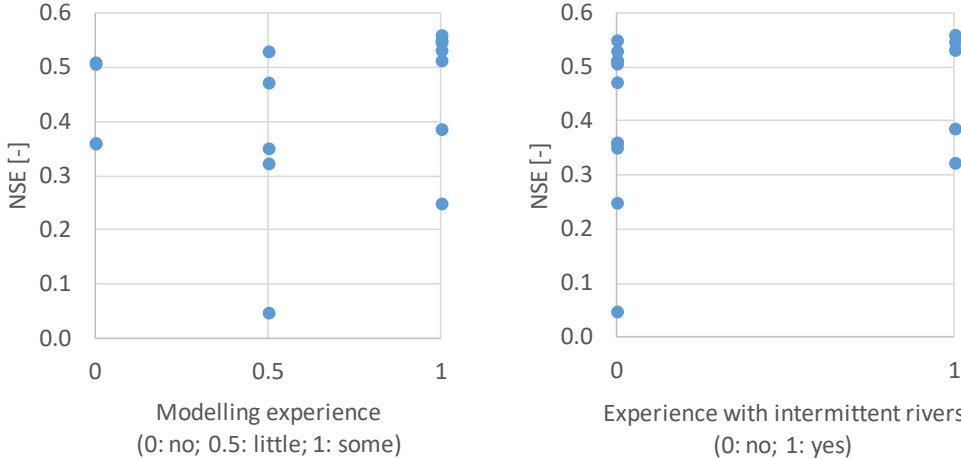

**Figure 5.** Dependence of model results (Nash-Sutcliffe model efficiency) on the modellers' experience in relation to model application (**Left**) and the knowledge of the specific characteristics of intermittent catchments (**Right**). For graphical reasons, the results of one modeller are not shown who achieved an NSE value of −45.9 (little modelling experience, no experience with intermittent rivers).

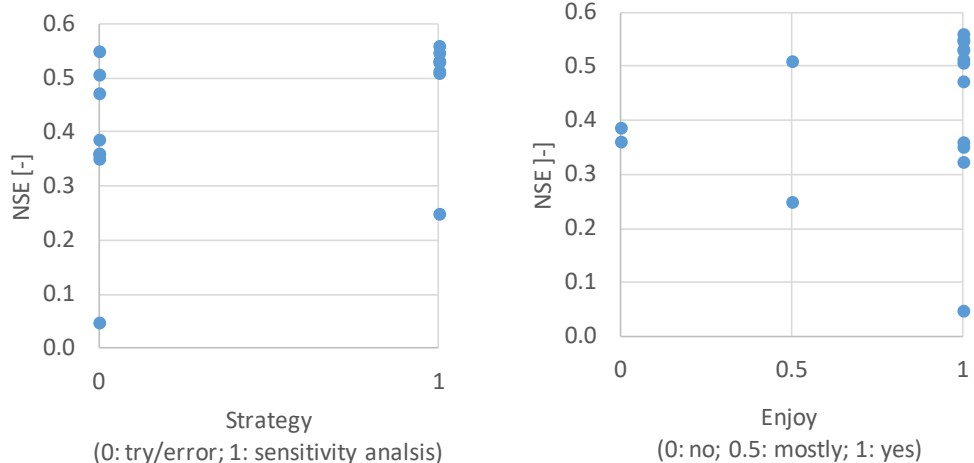

**Figure 6.** Dependence of model results (Nash-Sutcliffe model efficiency) on the calibration strategy (**Left**) and on whether the modellers' enjoyed the modelling contest or not (**Right**). For graphical reasons, the results of one modeller are not shown who achieved an NSE value of −45.9 (trial and error calibration, mostly enjoyed the contest).

Similar results were obtained from the question whether modellers enjoyed the contest or not. Although the variability in NSE values of those who enjoyed the contest was larger compared to those did not like it or criticized it, the best six model applications overall were obtained from those modellers who did enjoy the model application and the contest (Figure 6, right). Therefore, enjoyment seems to be an important factor for being successful for hydrological modellers.

Analysing multifunctional combinations could provide deeper insight, but the number of participants was much too small to come to a reliable quantitative estimation. Towards a multifunctional analysis, we conclude that the questionnaire should have been designed in a different way, asking for standardized answers concerning experience and attitudes. Additionally, such an experiment should be realized with a larger group of modellers to better represent the variability within the groups of modellers.

### 3.4. Improvement of the Model Results through Calculating Model(ler) Ensembles

Finally, the results of the modelling contest were used for falsifying the hypothesis "multi-modeller-ensembles cannot yield better results of conceptual catchment models than a single modeller can obtain due to the uncertainty of the modellers' decisions". For each day, the simulated discharges were used to calculate median- and mean-value-ensembles. Three different strategies were tested to select appropriate ensemble members:

(1) Considering all modellers: For this first multi-modeller-ensemble, the best simulated hydrograph (that with the highest NSE) simulated by each participating modeller was considered, independent of quality, strategy, or experience.

(2) Considering only those modellers who achieved satisfactory results (a posteriori): For this second multi-modeller-ensemble, only the best simulated hydrograph (that with the highest NSE) simulated by those participating modellers was considered who achieved satisfactory modelling results according to [25] (NSE > 0.5).

(3) Considering modellers according to their modelling experience (a priori): For this third strategy, the group of modellers was divided in two sub-groups, "participants with modelling experience" and "participants with little or without modelling experience". For those two sub-groups, ensembles were calculated based on the best simulated hydrograph (that with the highest NSE) simulated by the members of the sub-groups.

Comparing those model ensembles against all individual simulations, depending on the root mean squared error (RMSE) and the Nash-Suttcliffe efficiency (NSE), shows that:

(4) Independently of the quality of the individual model applications, those two ensembles based on all modellers' results have a simulation quality comparable to the best individual modeller (Figure 7).

(5) The two ensembles based on the simulations of those modellers' results who achieved "satisfactory" results even outperformed the best individual simulation and revealed the most robust simulation results (Figure 7).

(6) The two ensembles based on the simulated hydrographs of the experienced modellers also outperform all individual simulations (Figure 8) and even all other modellers ensembles investigated in this study (Figure 7). Additionally, both ensembles based on the simulated hydrographs of the less and unexperienced modellers achieved satisfactory modelling results according to [25] (NSE > 0.5).

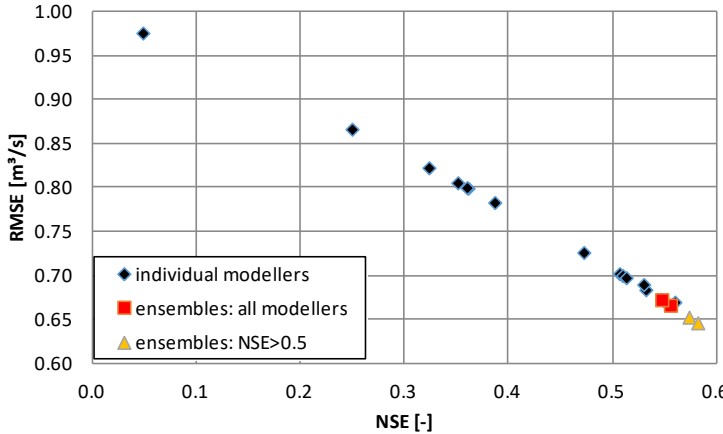

**Figure 7.** Comparison of individual model applications (blue diamonds) against model(er) ensembles (median, mean) based on all models (red squares) and on all models achieving satisfactory model results (yellow triangles) according to [25]. For graphical reasons, the results of one modeller are not shown who achieved an NSE value of −45.9.

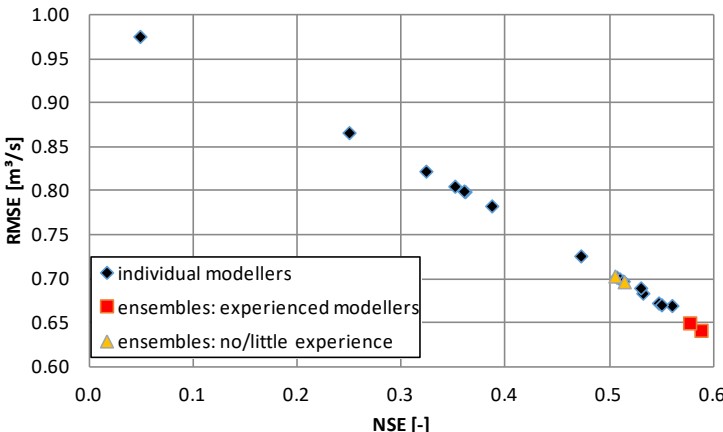

**Figure 8.** Comparison of individual model applications (blue diamonds) against model(er) ensembles (median, mean) based on the results of the experienced modellers (red squares) compared to the less or even unexperienced modellers (yellow triangles). For graphical reasons, the results of one modeller are not shown who achieved an NSE value of −45.9.

Hydrographs of selected multi-modellers ensembles are presented in Figure 9. In accordance with [14–16], the combination of individual simulations, therefore, increases the simulation quality, also in the case of different modellers applying the same catchment model. Thus, for the case of multi-modellers-ensembles in hydrological modelling, collective intelligence seems to be superior to individual intelligence.

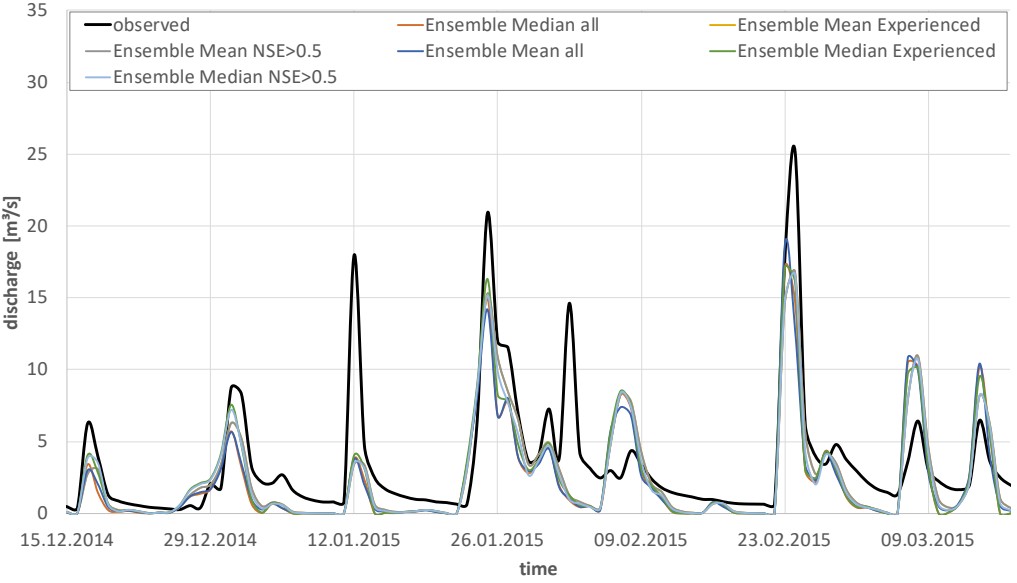

**Figure 9.** Variability of the discharges simulated by the investigated modellers' ensembles against observations for three exemplary months of the rainy season of 2014/2015.

## 4. Conclusions

The main findings from this modelling contest are:

(1)　One key requirement of successful application of a conceptual catchment model is that a modeller has a good calibration strategy. A systematic sensitivity analysis helps a lot to identify the most sensitive model parameters and to make better decisions in the (manual) calibration process.

(2)　Available experience in model application and knowledge on regional processes of course helps to achieve reasonable model results. However, it does not necessarily guarantee a high model performance.

(3)　To enjoy what one is doing—hydrological model application in this case—is a supporting factor. The analysis of the contest results confirmed this also for hydrological model application.

(4)　"Modeller ensembles" can be used to show the uncertainty caused by behavioural aspects. Combining model results of different modellers helps to make predictions more robust, also with regard to the modellers' decisions during the model application process. The good performance of a priori ensembles based on modellers' experience emphasizes the power of collective intelligence also for the case of hydrological model applications.

This study shows, therefore, that modellers' decisions and modellers' attitudes have a significant impact on the success of a hydrological model application. Apart from experience and regional process knowledge, strategic issues and enjoyment play an important role for a successful model application. Additionally, collective intelligence matters also in the case of hydrological modelling.

**Author Contributions:** H.B. and O.T. acquired the funding from DAAD; the study was designed by H.B., J.D. and O.T.; model simulations were delivered by H.B., M.M.d.B., D.C., D.C., A.D., P.F.G., J.K., A.K., P.K., A.K., N.K., J.M., V.M., M.M., K.M., C.S., and L.F.d.V.; data analysis and evaluation was done by H.B.; the original draft was prepared by H.B.; J.K., J.D. and O.T. assisted in reviewing and editing the draft.

**Funding:** This research was funded by DAAD (German Academic Exchange Service), grant "Floods and Flood Risk Management" as part of the program "Hochschuldialog mit Südeuropa".

**Conflicts of Interest:** The authors declare no conflict of interest.

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
