# Peer review of "Impact of Hydrological Modellers’ Decisions and Attitude on the Performance of a Calibrated Conceptual Catchment Model: Results from a ‘Modelling Contest’"

_hydrology, doi:10.3390/hydrology5040064_

Round 1
Reviewer 1 Report
Chap 1 Introduction
The introduction can and should be much shorter.
Try to really extract key finding by other authors. Avoid to mentioned authors at the begin of sentences like you did in line 75 to76.
The introduction ends with a list of the work done. However, it should end with clearly derived research questions of the study. Please reformulate the last paragraph (line 96-103) in this sense.
Chapter 2 Materials and methods
Is HBV a suitable model for the mediterranian catchment ? Where all data complete and seriously checked before being given to the participants ?
Can HBV be successfully calibrated for the selected catchment ? Which best results were derived by advanced calibration procedures (GLUE, stepped latin hypercube, optimization tools et al. ) ?
Only NSE and RSME were used as criteria for goodness of fit. Mass balances and peak flow deviations should be added as common basic criteria.
Where the participants allowed to communicate ?
It is not clear, whether and how the fitted parameters were checked on plausibility and admissibility.
The methodology is limited to investigating the influence of individual influencing factors on the calibration results. The study should incorporate (i) multifactorial influences and (ii) most successful combinations of factors resulting in best calibration results.
Chapter 3 Results and discussion
Chap 3.1: The uncertainty range of the flow data is missing. Fig 2 gives a good overview but is too small. Select characteristic events to show individual calibration effects.
Add information about parameter ranges of the calibrated parameters. Comment on their plausibility keeping in mind effects of equifinality.
How did the participants proceed during calibration. What was their individual strategy ? Document and evaluate the detailed strategies.
Fig 4 and Fig 5: space consuming. Results might be better given in one common table.
Fig 4: Consider if experience can be described more detailed (education in hydrology (CPs and grades), grades relevant courses et al.
Fig 5: Sensitivity analysis is NOT a calibration strategy. More details on the calibration strategy are necessary. Is joy a suitable criteria ? Might “engagement” be more suitable ?
Chap 3.4: Remember that strictly speaking, a hypothesis can only be falsified and not verified.
Fig. 6 and Fig 7: axes: Add dimensions
Show the hydrographs of the ensembles.
Author Response
Dear reviewer 1,
Thanks a lot for having provided valuable comments to our contribution. We have addressed most of them during the revision and hope that you will be satisfied by the adjustments made.
(1) The introduction can and should be much shorter.
In the introduction we try to elaborate the research question, to summarize the state of the art and to shortly introduce the way we try to address the research challenge and to bring thongs forward. We agree that the introduction should not be too extensive. But we could only significantly shorten the introduction if we either separated the state from the art from the introduction (which would not reduce the length in total) or if we removed parts of the description of the state of the art. In the submitted manuscript, this description in our opinion is not excessive. A reduction would lead to an incomplete description. We therefore just reformulated some sentences to slightly shorten the introductory text.
(2) Try to really extract key finding by other authors. Avoid to mentioned authors at the begin of sentences like you did in line 75 to76.
Sentences like line 75/76 were rephrased.
(3) The introduction ends with a list of the work done. However, it should end with clearly derived research questions of the study. Please reformulate the last paragraph (line 96-103) in this sense.
We reformulated the final paragraph of the introduction considering your advice.
(4) Is HBV a suitable model for the mediterranean catchment? Where all data complete and seriously checked before being given to the participants?
Literature analysis shows that HBV type model could be successfully applied to semiarid catchments. We therefore assume that it can also represent the hydrological processes of the selected Mediterranean catchment (a comment and further citations were added to the manuscript). Data were quality checked before the model applications (comment was added to the manuscript).
(5) Can HBV be successfully calibrated for the selected catchment ? Which best results were derived by advanced calibration procedures (GLUE, stepped latin hypercube, optimization tools et al. ) ?
This is a good point! It would be interesting to compare automatic calibration to manual calibration. However, optimization tools have not yet been applied to the data set, and a systematic analysis, consisting of a (systematic comparison of different tools for automatic calibration and manual calibration), would be beyond the scope of this study.
(6) Only NSE and RSME were used as criteria for goodness of fit. Mass balances and peak flow deviations should be added as common basic criteria.
As shown in section 3.1., in addition to Nash-Sutcliffe and RMSE also the coefficient of determination and the bias of the simulation were analysed. Therefore, mass balance was an important criteria. Peak flow deviations were not analysed in detail, but they are mainly governing the Nash-Sutcliffe model efficiency. We assume that peak flow deviation is sufficiently considered in the analysis.
(7) Where the participants allowed to communicate ?
The modellers were asked not to communicate on their individual model applications during the model contest. A comment on this was added to section 2.3.
(8) It is not clear, whether and how the fitted parameters were checked on plausibility and admissibility.
Plausibility of parameters was checked but not analysed in detail since most of the model parameters are of a conceptual nature. We added some information the parameter range used by the modellers (table 3) and commented on the problem of equifinality.
(9) The methodology is limited to investigating the influence of individual influencing factors on the calibration results. The study should incorporate (i) multifactorial influences and (ii) most successful combinations of factors resulting in best calibration results.
This is a good point. Unfortunately, the number of participants was limited to 17, and although we grouped the calibration strategy, attitude and experience in few groups based on the available information, the number of participants remains small compared to the number of possible combinations of influencing factors. Thus, analysing multifunctional combination could give some preliminary insight, but the number of participants would be much too small to come to any quantitative estimation. Therefore, we decided to stay with analysing individual factors and – by doing so – to draw conclusions based on a maximum number of participants.
(10) Chap 3.1: The uncertainty range of the flow data is missing. Fig 2 gives a good overview but is too small. Select characteristic events to show individual calibration effects.
The new figure 3 shows the variability of model predictions for three discharge events in winter 2015, as requested.
(11) Add information about parameter ranges of the calibrated parameters. Comment on their plausibility keeping in mind effects of equifinality.
Information on parameter ranges is provided in the new table 3, accompanied by a short discussion on equfinaility. We kept this short since data base (number of modellers) was not sufficient for a quantitative analysis.
(12) How did the participants proceed during calibration. What was their individual strategy ? Document and evaluate the detailed strategies.
Calibration strategy was only documented by answering the questions in the questionnaire. Most of the modellers calibrated their models based on trial and error. A smaller number executed a sensitivity analysis first to select the calibration parameters (see section 3.2). More details cannot be provided due to the design of the questionnaire.
(13) Fig 4 and Fig 5: space consuming. Results might be better given in one common table.
The reviewer is right, those 2 figure need space. However, they also give a clear impression on the variability of the model applications related to the modellers behaviour. Therefore we decided to keep the figures, but we rescaled them to save some space.
(14) Fig 4: Consider if experience can be described more detailed (education in hydrology (CPs and grades), grades relevant courses et al.
We discussed the scaling / grouping of experiences. Available information on experience was: number of models, number of years, experience with HBV, experience on semiarid regions. Finally, to keep the number of modellers reasonable, we decided to keep the groups. We think that our conclusions based on that approach are justified.
(15) Fig 5: Sensitivity analysis is NOT a calibration strategy. More details on the calibration strategy are necessary. Is joy a suitable criteria ? Might “engagement” be more suitable ?
The referee is correct. We adjusted our formulations and wording to account for this correction. Some more explanation on the calibration is provided in section 3.2. With regard to the criteria, “joy” of course can be discussed. In the questionnaire we explicitly asked whether the participants ”enjoyed” the contest. That’s why we used that expression.
(16) Chap 3.4: Remember that strictly speaking, a hypothesis can only be falsified and not verified.
Thanks for this advice. We adjusted the wording.
(17) Fig. 6 and Fig 7: axes: Add dimensions
Dimensions were added.
(18) Show the hydrographs of the ensembles.
A hydrograph of the ensembles was added for an exemplary time period of the rainy season (new figure 9).
Reviewer 2 Report
Review Comments for Impact of hydrological modellers’ decisions and attitude on the performance of a calibrated conceptual catchment model: results from a ‘modelling contest’(hydrology-383837) The authors raised a very interesting question in this manuscript (MS): how do hydrological modellers’ decisions and attitude affect the performance of the HBV model. 17 modelers from summer school provided a unique chance to address the question. Overall, I think the MS is interesting and can be considered for publication in Hydrology if my following concerns can be addressed. Major issues: (1) Only 3 years data were used for analysis, which, from the perspective of hydrology, is not long enough. Moreover, the first year is not preferable for analysis in many cases considering the uncertainty of initial conditions. The authors should clarify this issue or at least add some discussions. (2) Figure 2 clearly shows that none of the modeler can reproduce the recession period of several major floods. I think it is related to the parameters of recession rate of the various reservoirs (not sure, just based on my experience of other models). My question here is did modelers well trained for calibrating the model? (3) All results are from students, why not also add the results from teacher- a well-trained modeler. Minor Issues: (1) Section 3.2, better to present these data in tables or figures. (2) The title of Section 2.3 and 2.4 are same. (3) Figure 6 and 7, RMSE should has unit. (4) Figure 1 is unnecessary, better to replace it with schematic figure of the study area.Author Response
Dear reviewer 2,
Thanks a lot for having provided valuable comments to our contribution. We have addressed most of them during the revision and hope that you will be satisfied by the adjustments made.
(1) Only 3 years data were used for analysis, which, from the perspective of hydrology, is not long enough. Moreover, the first year is not preferable for analysis in many cases considering the uncertainty of initial conditions. The authors should clarify this issue or at least add some discussions.
It is correct that usually more than 3 years data should be used to capture the variability of the catchment hydrological system. However, in this case high data quality could only be guaranteed for the 3 years period. For that time the local University ran an own observational system and could check the data for quality. The uncertainty of the initial conditions was reduced by starting the experiment in the end of the dry season in 2014, assuming that dry moisture conditions are justified. Some information was added at the end of section 2.2.
(2) Figure 2 clearly shows that none of the modeller can reproduce the recession period of several major floods. I think it is related to the parameters of recession rate of the various reservoirs (not sure, just based on my experience of other models). My question here is did modellers well trained for calibrating the model?
As mentioned in the manuscript, the model was introduced prior to the modelling contest. Process equations and parameterization was explained. (A more detailed comment on this was added to section 2.3 in the manuscript). Whether the modellers are well trained or not, should be part of the result of the experiment! We assume that experienced modellers should do a better job in calibration. In general this assumption applied, but not for the recession curve. However, you are right that none of the modeller represented the recession curve right. This could be for example due to insufficient time of the experiment or specific catchment conditions (see also comment in section 3.1), but diving deeper into this would be speculation.
(3) All results are from students, why not also add the results from teacher- a well-trained modeller.
MSc students, PhD students and Postdocs with diverse modelling experience were participating in the modelling experiment. One of them is a teacher in hydrological modelling. May be this was not clear in the submitted version of the manuscript. We added some information to clarify this aspect.
(4) Section 3.2, better to present these data in tables or figures.
Reviewer 1 addressed the issue if space consumption of tables and figures. That’s why we desisted from adding new tables and figures here, since several tables and/or figures would be necessary to illustrate the data presented in the text.
(5) The title of Section 2.3 and 2.4 are same.
The title of section 2.4 was adjusted.
(6) Figure 6 and 7, RMSE should has unit.
Units were added to RSME in Figures 6 and 7.
(7) Figure 1 is unnecessary, better to replace it with schematic figure of the study area.
Figure 1 (HBV model) was removed, a figure on the catchment was added (new figure 1)).
Round 2
Reviewer 1 Report
Point (3): The new text doesn´t meet the commented point. Simply list your basic research questions which base on the above mentioned summary of the state of research and which will be able to contribute to the proceed of knowledge. You can keep it short and simple: question 1, question 2,… Name the experiment you choosed to get answers from. Details are given in Chap 2.
Point (8): Table 3: x and k obviously had been set constant by all participants. Why ? Remove them from table 3 as on value cannot be min, med and max at the same time.
Point (9): Include main points of your comment into the text
Point (12): Include main points of your comment into the text
Point (14): reconsider your comment. You have more detailed data about the information which get lost by aggregation which is not necessary. The argument that the small number of participant is not convincing. A more detailed analysis may come up with a new hypothesis for a new research set up. Pleas reconsider this point.
Point (16): The wording chosen does not meet the point. You cannot test a hypothesis “that …” or “whether…” . You can only falsify a hypothesis. In your case the hypothesis to be falsified is “multi-modeller-ensemble can not yield better results …”. Considering the results given in Fig 9 (excellent!) the test result is “not falsified”. It seems to but tricky but that is the way science works ..
Good luck with your paper and inspiring future research work.
Author Response
Dear reviewer 1,
Thanks a lot, once more, for having provided valuable comments to our contribution. We have addressed most of them during the revision and hope that you will be satisfied by the adjustments made.
Point (3): The new text doesn´t meet the commented point. Simply list your basic research questions which base on the above mentioned summary of the state of research and which will be able to contribute to the proceed of knowledge. You can keep it short and simple: question 1, question 2,… Name the experiment you choosed to get answers from. Details are given in Chap 2.
We have added three research questions at the end of the introduction according to your suggestion.
Point (8): Table 3: x and k obviously had been set constant by all participants. Why ? Remove them from table 3 as on value cannot be min, med and max at the same time.
X and k were not touched by the modellers. They mainly focused on the adjustment of soil parameters and tried to calibrate stream flow by varying recession constants and the routing parameter. Therefore we removed the values from table 3 as suggested.
Point (9): Include main points of your comment into the text
We included the main points into the text (section 3.3)
Point (12): Include main points of your comment into the text
Main points are included in the text (section 2.4 in the questionnaire, and section 3.2. in the evaluation of the results of the questionnaire).
Point (14): reconsider your comment. You have more detailed data about the information which get lost by aggregation which is not necessary. The argument that the small number of participant is not convincing. A more detailed analysis may come up with a new hypothesis for a new research set up. Pleas reconsider this point.
We did not have such information you asked for in your first review (e.g., education in hydrology (CPs and grades), grades relevant courses et al.). Concerning modelling experience, we asked three questions on: (1) number of models, (2) number of years, (3) experience with HBV. Only one person had experience with HBV but did not significantly vary from the others. So 2 numbers were remaining. To work with groups and not with individual modellers, we decided to build classes (see section 2.4). For a more detailed analysis we think that we should have designed the questionnaire in a different way asking for standardized answers concerning experience, and we recommend to work with larger groups of modellers to better represent the variability within the group of modellers (commented on in section 3.3).
Point (16): The wording chosen does not meet the point. You cannot test a hypothesis “that …” or “whether…” . You can only falsify a hypothesis. In your case the hypothesis to be falsified is “multi-modeller-ensemble can not yield better results …”. Considering the results given in Fig 9 (excellent!) the test result is “not falsified”. It seems to but tricky but that is the way science works ..
We did adjust the wording once more according to your suggestion. Thanks for your advice!
Reviewer 2 Report
I think the current version can be considered for publication in Hydrology.
Author Response
Dear reviewer 2,
thanks once more for your valuable comments and for the approval!